


# A Single Parameter Hygroscopicity Model for Functionalized and Insoluble Aerosol Surfaces

**Chun-Ning Mao[1], Kanishk Gohil[1] and Akua Asa-Awuku[1,2]**

[1] Department of Chemical and Biomolecular Engineering, University of Maryland, College Park, MD 20742 USA

[2] Department of Chemistry and Biochemistry, University of Maryland, College Park, MD 20742 USA

*Correspondence to*: Akua A. Asa-Awuku (asaawuku@umd.edu)

**Abstract.** The impact of molecular level surface chemistry for aerosol water-uptake and droplet growth is not well understood. In this work, spherical, non-porous, monodisperse Polystyrene Latex particles treated with different surface functional groups are exploited to isolate the effects of aerosol surface chemistry for droplet activation. PSL is effectively water-insoluble and changes in the particle surface may be considered a critical factor in the initial water-uptake of water insoluble material. The droplet growth of two surface modified types of PSL (PSL-NH$_2$ and PSL-COOH) along with plain PSL was measured in a supersaturated environment with a Cloud Condensation Nuclei Counter (CCNC). Three droplet growth models - traditional Köhler (TK), Flory-Huggins Köhler (FHK) and the Frenkel-Halsey-Hill adsorption theory (FHH-AT) were compared to experimental data. The experimentally determined single hygroscopicity parameter, $\kappa$, was found in the range from 0.002 to 0.04. The traditional Köhler prediction assumes Raoult's law solute dissolution and underestimates the water-uptake ability of the PSL particles. FHK can be applied to polymeric aerosol; however, FHK assumes the polymer is soluble and hydrophilic. Thus, the FHK model generates a negative result for hydrophobic PSL and predicts non-activation behavior that disagrees with the experimental observation. The FHH-AT model assumes that a particle is water-insoluble and can be fit with 2 empirical parameters ($A_{FHH}$ and $B_{FHH}$). The FHH-AT prediction agrees with the experimental data and can differentiate the water uptake behavior of the particles due to surface modification of PSL surface. PSL-NH$_2$ exhibits slightly higher hygroscopicity than the PSL-COOH, while plain PSL is the least hygroscopic among the three. This result is consistent with the polarity of surface functional groups and their affinity to water molecules. Thus, changes in $A_{FHH}$ and $B_{FHH}$ can be quantified when surface modification is isolated for the study of water-uptake. The fitted $A_{FHH}$ for PSL-NH$_2$, PSL-COOH and plain PSL is 0.23, 0.21 and 0.18 when $B_{FHH}$ is unity. To simplify the use of FHH-AT for use in cloud activation models, we also present and test a new single parameter framework for insoluble compounds, $\kappa_{FHH}$. $\kappa_{FHH}$ is within 5% agreement of the experimental data and can be applied to describe a single-parameter hygroscopicity for water-insoluble aerosol with surface modified properties.

## 1. Introduction.

Heterogeneous water-vapor condensation occurs for both soluble (Rose et al., 2008) and insoluble (Dalirian et al., 2018; Koehler et al., 2009; Kumar et al., 2009) particles. Traditionally, the cloud condensation nuclei (CCN) activation behavior is described by Köhler theory (Köhler, 1936). In traditional Köhler (TK) theory, the droplet is assumed to be dilute, and the water activity follows Raoult's law; such that the water activity equals the water mole fraction, and the dilute droplet water activity coefficient is assumed to be unity. For water-soluble particles like inorganic ammonium sulfate (Rose et al., 2008) and sucrose (Dawson et al., 2020; Gohil and Asa-Awuku, 2022), TK can accurately predict their water uptake behavior. However, TK does not work so well for atmospherically relevant and abundant particles that are partially water soluble or





water insoluble (Kumar et al., 2009; Tang et al., 2016).  Thus, alternative droplet growth models for the
partial and insoluble particles are needed.

The droplet formation of water-insoluble particles has been previously described with adsorption
activation models (Hatch et al., 2012; Kumar et al., 2009; Lintis et al., 2021; Malek et al., 2022; Navea et al.,
2017; Pajunoja et al., 2015; Tang et al., 2016). Brunauer, Emmett, and Teller (BET) adsorption isotherm
models are typically applied for multilayer adsorption analysis of water uptake on clays (Hatch et al., 2012)
and fly ash (Navea et al., 2017). Lintis et al., 2021 applied the Dubinin-Serpinsky model for soot and
concluded that nucleation occurred at oxidized and hydrophilic surface sites on the soot. Pajunoja et al.
measured the TK hygroscopicity for SOA and demonstrated that the droplet growth of organic compounds
with low O:C ratio was an adsorption-dominated process for both sub – and super-saturated water vapor
conditions. Kumar et al. 2009 showed that the droplet activation of mineral dust could be described well
using the Frenkel-Halsey-Hill adsorption theory (FHH-AT).

FHH-AT determines droplet growth with the help of 2 empirical parameters defined as $A_{FHH}$ and $B_{FHH}$
(Sorjamaa and Laaksonen, 2007a). The parameter $A_{FHH}$ is related to the interaction of the first layer of water
and the particle surface, while the $B_{FHH}$ represents the interaction between other layers of water molecules
and the particles.  Furthermore, it is postulated that $A_{FHH}$ should range from 0.1 to 3.0, and $B_{FHH}$ should be
within range of 0.5 to 3.0 for mineral dust aerosols (Kumar et al., 2009).  However, the reported and applied
$A_{FHH}$ and $B_{FHH,}$ parameters have varied significantly in literature. For example, Karydis et al. 2017 used the
FHH-AT model to simulate the aerosol indirect effect of insoluble mineral dust CCN with an $A_{FHH}$ and $B_{FHH}$
of 2.25±0.75, and 1.20±0.10 respectively. Furthermore, Karydis et al., 2017 concluded that the global CCN
number would decrease 10% for mineral dust with the use of those values. Hatch et al., 2014 measured
montmorillonite dust and found that the $A_{FHH}$ and $B_{FHH}$ was $98 \pm 22$ and $1.79 \pm 0.11$ and the $A_{FHH}$ and $B_{FHH}$
was $75 \pm 17$ and $1.77 \pm 0.11$ for illite, respectively. Particles of intermediate polarity and high porosity may
lead to a higher CCN activity (Koehler et al., 2009) and Hatch et al., 2012 attributed the high value of $A_{FHH}$
to the surface chemistry and the porosity of the clays.

Specifically, porosity leads to a higher perceived hygroscopicity because water molecules fill the
microporous structure. Thus, it should be noted that in the aforementioned studies, the porosity of insoluble
particles complicates the study of aerosol water-uptake and influences the role of aerosol chemistry that
impacts the CCN activity. Moreover, the use of two parameters adds an additional degree of freedom that
makes the direct comparison of different chemical species challenging; there may be multiple pairs $A_{FHH}$ and
$B_{FHH}$ solutions for a single compound with or without porous structure. As a result, direct experimental
measurement of the hygroscopicity of non-porous and spherical nuclei is important to understand the
adsorption water uptake ability of atmospheric aerosol.

To better understand the activation behavior of insoluble particles, we study the CCN activity of
spherical, non-porous polystyrene latex (PSL) particles using the Cloud Condensational Nuclei Counter
(CCNC) (Roberts and Nenes, 2005). PSL is a common material used for instrumentation calibration and
model examination for aerosol optical properties. (Miles et al., 2011; Pettersson et al., 2004; Singh et al.,
2014). They are spherical, uniform in size, and do not dissolve in water. The surfaces of PSL particles are
hydrophobic, but can be manufactured to have a high density of hydrophilic sites (Ottewill and Vincent,
1972). Different functional groups like carboxyl (-COOH) or amine ($-NH_2$) groups can be added to the PSL
surface to create hydrophilic adsorption sites. Previous studies have shown that surface chemistry of the



insoluble particles affect ice nucleation (Cziczo et al., 2009; Koehler et al., 2009; Reitz et al., 2011; Sullivan et al., 2010). But little is known about the influence of surface chemistry for liquid cloud droplet activation.

To our knowledge, few studies have explored the effects of molecular level particle surface changes for CCN and droplet formation. Mainly, water-soluble aerosol that contribute solute to the solution and modify droplet properties has been of great interest. Any changes at the surface of soluble aerosol are often overshadowed by the effects of solute dissolution if the solute is water-soluble. Additionally, to understand impacts of functionalized surfaces, the surface area or particle size of an aerosol must be well known. Given these constraints, spherical PSL aerosol with and without a functionalized surface provides an opportunity to advance the contributions of molecular surfaces to the discussion of water-uptake.

In the following sections, we examine the impact of surface chemistry to the CCN activity in both the Traditional Köhler (TK), Flory Huggins Köhler (FHK) and the FHH-AT model. The first two models are generally applied to water-soluble compounds while the FHH-AT is for water-insoluble particles. Furthermore, FHK is specifically designed for high-molecular weight polymers like PSL and therefore briefly considered. The following work also compares measurements to three hygroscopicity prediction models. Two single parameter hygroscopicity representations have been previously derived using traditional Köhler Theory (Petters and Kreidenweis, 2007) and Flory Huggins Köhler (Mao et al., 2021) assumptions. Additionally, we derive a third theoretical hygroscopicity parameter using the FHH-AT model and analyze the role of surface chemistry in the adsorption based hygroscopicity. Thus, the following data and analysis provides insight into the water-uptake of water-insoluble particles and the impact of surface modified functional groups for the perceived aerosol hygroscopicity and droplet formation.

## 2. Experimental Procedure.

### 2.1 Polystyrene Latex (PSL) Composition and Size

Four different particle sizes, ~100nm, 200nm, 300nm and 500nm and three different PSL particles (plain, surface modified with amine functional group, $-NH_2$, and carboxyl functional group, $-COOH$) were purchased (Lab 261®). The sizes provided by the manufacturer are determined with dynamic light scattering (DLS) techniques. In this study, we verify the size and report the electrical mobility measured particle size ($D_d$) of the PSL particles with a Differential Mobility Analyzer, DMA (TSI 3080) and Condensation Particle Sizer (TSI 3776) operated in size scanning mode (Wang and Flagan, 1990). The measured geometric mean size was within ~10% difference of the manufacturer's reported particle sizes (Table 1).

### 2.2 Aerosol Generation

0.4 ml of the PSL particle solution was diluted in 50ml ultra-purified water (Millipore®, with conductivity < 18.8MΩ). Wet particles were then generated with a constant output atomizer (TSI, 3076). Wet droplets were then passed through two silicone dryers and the relative humidity was 5% after passing through the dryers. After drying, large particles were removed by a 0.71 cm impactor to prevent the multiple charging errors. Poly-disperse particles are charged and sampled by an electrostatic size classifier, specifically a Differential Mobility Analyzer (TSI, DMA 3080). The DMA was set to select the size of the PSL particles. The PSL particles are then passed through a Condensation Particle Counter (CPC, TSI 3776) with a flow rate





of 5 cm$^3$ s$^{-1}$ and a CCNC. The particle density counted by the CPC is the condensation number concentration
(CN) and is measured at rate of 1 Hz. The sheath and sample flow ratio are 10:1 in both the CPC and CCNC.

**2.3 The Critical Supersaturation of PSL.**

The CCN activity for the selected particle size was measured with a continuous flow stream-wise thermal gradient Cloud Condensation Nuclei Counter (DMT CCN100) (Roberts and Nenes, 2005). A brief introduction is provided here and readers are directed to the (Roberts and Nenes, 2005) for a more detailed
discussion of the instrument. The CCNC is a column with a wet inner surface. Three thermal electrical controllers modify temperatures on the top, in the middle, and at the bottom of the CCNC column to establish a constant temperature gradient. A supersaturation is generated at the center of the column as air moves from the top to the bottom (Hoppel et al., 1979). The sampled particles in the CCNC column provide surface for the occurrence of the heterogeneous condensation. An optical particle counter (OPC) at the bottom of the
column, counts the particles that form droplets greater than 0.75 μm and provides the cloud condensation number concentration (CCN) every 1 Hz. Instrument supersaturations were achieved by modifying both inlet flowrate and the temperature gradient in the CCNC. Each supersaturation was calibrated using ammonium sulfate and Scanning Mobility CCN Analysis (SMCA) (Moore et al., 2010).

CCN activity is the ratio of number droplets to total aerosol (CCN/CN) measured at a given
supersaturation and constant particle size ($D_d$). For each supersaturation, temperatures and flows are held constant for 10 minutes and CCN and CN data are measured every second. The CCN and CN concentration is then averaged in the 8$^{th}$ minute. The CCN activation for each sample is reported from 0.1 to 1.4% supersaturation. Critical supersaturation ($s_c$) for a given particle size ($D_d$) is defined at 50% efficiency growth of the CCN (i.e., CCN/CN=0.5). A smaller $s_c$ for a constant $D_d$ indicates that the particles are more
hygroscopic. The $s_c$ for all twelve PSL are listed in Table 1. The activation curves for all twelve PSL samples are provided in Supplemental Materials (Figure S-1). The measured $s_c$ and $D_d$ values are used to compute and compare subsequent particle CCN activation and hygroscopicity.

**3. Theory and Analysis**

The saturation ratio at the droplet surface can be generally described as follows.

$$S = a_w \exp\left(\frac{A}{D}\right); \text{ and } A = \frac{4 M_w \sigma_w}{R T \rho_w} \qquad , \qquad (1)$$

where $S$ is the saturation ratio; $a_w$ is the water activity of the solution and $D$ is the wet diameter of the droplet. The exponential term is known as the Kelvin term and describes the homogeneous nucleation of the droplet solvent. Thus, $A$ is generally constant and is a function of the universal gas constant ($R$), the pure water droplet surface tension ($\sigma_w$), temperature ($T$), density ($\rho_w$) and molecular weight ($M_w$).
Heterogeneous nucleation is considered in the water activity term. As aforementioned, for water-soluble inorganic salts and organics, (e.g., ammonium sulfate and sucrose) the water activity is approximated with Raoult's law and the water mole fraction, $x_w$.

Petters and Kreidenweis, 2007 employed Raoult's law to develop a single hygroscopicity parameter representation, $\kappa$, as follows,

$$\frac{1}{a_w} = 1 + \kappa \frac{v_s}{v_w}, \qquad (2)$$





Where $v_s$ is the total volume of the dry particle and $v_w$ is the total volume of the water in a droplet. The $\kappa$ in (2) is defined as the intrinsic hygroscopicity parameter of the compound, denoted by $\kappa_{int}$. If one assumes a dilute droplet, such that $a_w = 1$, $\kappa_{int}$ can be solved from known solute and solvent properties such that $\kappa_{int} = \frac{M_w \rho_s}{M_s \rho_w}$ (Sullivan et al., 2009). Where $M_w$ is the molecular weight of water; $M_s$ is the molecular

weight of the dry particle; $\rho_w$ is the density of water; $\rho_s$ is the density of the dry particle. If PSL is a polymer with ~100,000 g mol$^{-1}$ molecular weight and a density of 1.06g cm$^{-3}$, $\kappa_{int} \sim 0.0002$, is a small number and approaches zero. One can also derive a hygroscopicity parameter based on TK, $\kappa_{TK}$, directly from measured experimental $s_c$ and $D_d$ data (Petters and Kreidenweis, 2007)

$$\kappa_{TK} = \frac{4A^3}{27 D_d^3 ln^2 s_c},$$ (3)

$\kappa_{int} = \kappa_{TK} = 0.604$ for ammonium sulfate (Rose et al., 2008). The intrinsic and experimentally derived values also agree well for water-soluble compounds and partially soluble organics (Dawson et al., 2020; Peng et al., 2021, 2022).

Traditional Köhler theory calculations from theory and measurement tend to disagree for high molecular weight organics, such as polymers (Petters et al., 2006, 2009). To calculate the water activity of

high molecular weight compounds, Petters et al., 2006, 2009 combined the water activity of Flory-Huggins(Flory, 1942) with the Köhler theory for polymeric aerosols. Mao et al., 2021 derived a single parameter based hygroscopicity representation, $\kappa_{FHK}$ as follows,

$$\kappa_{FHK} = \frac{1-\varphi}{\varphi}\left[-1 + \frac{1}{(1-\varphi)\,exp[(1-F)\varphi + \chi\varphi^2]}\right],$$ (4)

where $\varphi$ is the volume fraction of the polymer. $F$ is the reciprocal of the chain segments of the polymer

equal to the ratio of the molecular volume of water and the solute and $\chi$ is the Flory-Huggins interaction parameter. Measured particle diameter, $D_d$, and $s_c$ data are used to define empirical fits of $\chi$ and subsequently determine, $\kappa_{FHK}$.

Insoluble aerosol droplet activation is best described by an adsorption thermodynamic droplet growth model (Kumar et al., 2009). Sorjamaa and Laaksonen, 2007a suggested the Frenkel-Halsey-Hill adsorption

theory (FHH-AT) to define the $a_w$ using the isotherm as follows,

$$a_w = \exp\left(-A_{FHH}\theta^{-B_{FHH}}\right) \quad \text{and} \quad \theta = \frac{D - D_d}{2D_w},$$ (5)

where $\theta$ is the surface coverage, and describes the layers of water molecules adsorbed on to the dry particle surface (Sorjamaa and Laaksonen, 2007b). $D_w$ is the diameter of a single water molecule and equals to 0.275 nm. $A_{FHH}$ and $B_{FHH}$ are compound specific empirical parameters. The FHH-AT parameters can be

estimated by fitting the FHH-AT with $s_c$ and $D_d$ CCN measurement data (Herich et al., 2009; Kumar et al., 2009).

To date, a single parameter hygroscopicity representation based on adsorption droplet growth does not exist in the current literature. In this work, we derive an experimental and theoretical adsorption hygroscopicity, $\kappa_{FHH,exp}$ and $\kappa_{FHH,the}$ respectively. If the critical wet droplet diameter, $D_{p,c}$, is known,

$\kappa_{FHH,exp}$, can then be expressed as a function of $D_d$ and $D_{p,c}$ as follows





$$\kappa_{FHH,exp} = f(D_d, D_{p,c}) = \frac{6\theta D_w}{D_d}\left(\frac{1}{\exp(-A_{FHH}\theta^{-B_{FHH}})} - 1\right), \tag{6}$$

Theoretically, droplet activation occurs when the derivative of the Köhler curve equals to zero $\left(\frac{dS}{dD_{p,c}} = 0\right)$. Hence the relation between critical surface coverage and the dry particle size is constrained with the following equation,

$$1 - \frac{2\theta_c D_w}{D_d} = \left(\frac{2AD_w}{A_{FHH}B_{FHH}D_d^2}\right)^{1/2}\theta_c^{\frac{B_{FHH}+1}{2}}, \tag{7}$$

The critical surface coverage, $\theta_c$, is defined at the point where droplet activation occurs, and is obtained by solving Eq. (7). As a result, the theoretical hygroscopicity of FHH-AT is derived,

$$\kappa_{FHH,the} = f(D_d) = \frac{6D_w}{D_d}A_{FHH}\theta_c^{-B_{FHH}+1}, \tag{8}$$

Readers should refer to Appendix A for additional derivation details.


## 4. Results and Discussion

### 4.1 The CCN activity for different types of PSL.

Table 1 shows the mobility diameter of PSL and their corresponding critical supersaturation. Particle size matters most for water-uptake and thus larger particles (~500nm) exposed to a constant supersaturation
activate earlier than smaller aerosol (~100nm) (Dusek et al., 2006). For example, PSL-NH2 particles with 85 nm diameters activate at 1.42% supersaturation while particles with 375nm activate at 0.33% supersaturation. PSL-COOH and the plain PSL show a similar trend as well. The data suggests that PSL particles are wettable and hygroscopic, more so than particles with $\kappa = 0$. The parameters in Table 1 are used to predict CCN activation ($s_c$-$D_d$ pairs) in Figure 1(a) and (b). The red dashed line in Figure 1 shows the traditional Köhler
prediction for ammonium sulfate ($\kappa_{int} = 0.604$) for comparison purposes. All PSL particles are much less hygroscopic than ammonium sulfate ($\kappa_{int} = 0.604$) with larger hygroscopicity than theoretical values derived from known solute properties ($\kappa_{int} = 0.0002$). Traditional Köhler theory significantly underpredicts particle activation and droplet growth of PSL particles.

Table 1 also shows the fitted parameters for the FHK, and FHH-AT models required to subsequently
calculate $\kappa_{FHK}$ and $\kappa_{FHH,the}$. Both FHK and FHH-AT models have additional degrees of freedom compared to the traditional Köhler theory. In the FHK model, the molar volume becomes negligible and the water-polymer interaction parameter ($\chi$), drives the droplet activation. $\chi$ describes the repulsive and attractive force between the solvent and the polymer. A $\chi$ smaller than 0.5 is an indication of miscibility and that water is a "*good solvent*" (Pethrick, 2004). $\chi$ is the only empirical free parameter in this study and when fitted to all
three types of PSL, $\chi > 0.5$ and confirms the assumption that PSL particles are water insoluble. In Figure 1a, the FHK model with only one free fitting parameter more closely agrees with experimental data than the TK theory (red solid lines).

Two empirical parameters ($A_{FHH}$ and $B_{FHH}$) for the FHH-AT model are reported for each type of the PSL. The FHH-AT model with two degrees of freedom agrees with the experimental data better than the


traditional Köhler theory model (Figure 1b). However, multiple solutions of $A_{FHH}$ and $B_{FHH}$ may exist. The fit results for plain type PSL estimate $A_{FHH} = 0.17$ and $B_{FHH} = 0.99$; $A_{FHH}$ is 0.3 and $B_{FHH}$ is 1.08 for carboxyl functional group modified PSL and $A_{FHH}$ is 0.11 and $B_{FHH}$ is 0.83 for amine functional group modified PSL. Differences in the $A_{FHH}$ and $B_{FHH}$ values confirm that the FHH-AT model is sensitive to molecular level chemistry and can distinguish changes in surface chemistry. PSL-COOH shows a higher attraction to water

molecules than the plain PSL. However, PSL-NH$_2$ is the most hygroscopic among the three types of the PSL and the $A_{FHH} = 0.11$ and the $B_{FHH} = 0.83$ of PSL-NH$_2$ are the lowest values among all three types. If $A_{FHH}$ represents the interaction between the first layer water molecules and the surface of the PSL particles, a higher $A_{FHH}$ value implies higher attractive forces. Thus, the unconstrained FHH-AT parameter solutions must be reassessed.

The only chemical difference between the three types of PSL is due to surface modification. The attraction force between the particle core and the layers of the water molecules should be the same. Thus, a second constrained best-fit solution exists if we restrict $B_{FHH} = 1$ (Table 1). The constrained and fitted $A_{FHH}$ is 0.18 for plain PSL, 0.21 for PSL-COOH and 0.23 for PSL-NH$_2$ (Table 1, Figure 1b). With the constrained solution, the higher $A_{FHH}$ is consistent with the most hygroscopic aerosol species. In Figure 1b, the FHH-AT

model prediction agrees well with the experimentally measured $s_c$-$D_d$ data more than traditional Kohler Theory. This agreement with data is true for both the best-fit constrained ($B_{FHH} = 1$, Figure 1b) and unconstrained $A_{FHH}$ and $B_{FHH}$ values (not shown).

Table 1. Important aerosol physical properties and parameters used to derive droplet growth.

| $\kappa_{int}$ (-) | $\chi$ (-) | $A_{FHH}$(-) | $B_{FHH}$(-) | Surface Modification | $D_d$ (nm) | $s_c$ (%) |
|---|---|---|---|---|---|---|
| 0.0002 | 0.56 | 0.18 | 1 | Plain | 85 | 1.27 |
| | | | | | 250 | 0.43 |
| | | | | | 310 | 0.42 |
| | | | | | 474 | 0.33 |
| 0.0002 | 0.54 | 0.21 | 1 | -COOH Carboxyl | 89 | 1.08 |
| | | | | | 223 | 0.61 |
| | | | | | 331 | 0.36 |
| | | | | | 472 | 0.25 |
| 0.0002 | 0.57 | 0.23 | 1 | -NH$_2$ Amine | 85 | 1.42 |
| | | | | | 195 | 0.63 |
| | | | | | 278 | 0.35 |
| | | | | | 375 | 0.33 |



### 4.2 The Impact of Surface Chemistry to Hygroscopicity of PSL

The perceived single parameter hygroscopicity, $\kappa_{TK}$ and the $\kappa_{FHH}$ can be derived from traditional Köhler and FHH-AT models, respectively (Figure 2). In TK, the theoretical hygroscopicity is a constant value independent to the size of the particles. In Figure 2 (a), the red solid horizontal line shows $\kappa_{int} = 0.0002$. The red open symbols (circle, square and triangle) are the hygroscopicity from measured data (Equation 3) of the respective PSL particles. The experimental $\kappa_{TK}$ values from traditional Köhler theory range from 0.002 to 0.04. Additionally, $\kappa_{TK}$ of the PSL are size dependant and are larger than $\kappa_{int}$. Thus, TK should not be applied to predict the droplet growth and single parameter hygroscopicity of insoluble PSL particles.

In addition, FHK should not be used to predict droplet growth of PSL. Although the FHK model agrees well with $s_c$-$D_d$ data (Figure 1a), the derived hygroscopicity derived is problematic and nonsensical (therefore not shown). FHK assumes that the hydrophilic polymer swells in the water droplet and the interaction parameter represents the molecular force between water and polymer. $\chi$ values larger than 0.5 derived from PSL-NH$_2$, PSL-COOH and plain PSL subsequently estimate negative and implausible $\kappa_{FHK}$ values. The experimental data indicate that the PSL indeed grow into droplets and have a positive hygroscopicity. Thus, FHK should not be applied to water-insoluble polymers like PSL.

The hygroscopicity values derived from the FHH-AT model are plausible and show the best agreement between theory and experiment (Figure 2). Both theoretical ($\kappa_{FHH,the}$) and experimental ($\kappa_{FHH,exp}$) hygroscopicity values from FHH-AT model are less than the $\kappa_{TK}$ (red open symbols), but higher than a constant $\kappa_{int}$ value of 0.0002. $\kappa_{FHH,exp}$ and $\kappa_{FHH,the}$ for all twelve PSL are within 5% of each other. The small deviation between $\kappa_{FHH,the}$ and $\kappa_{FHH,exp}$ demonstrate that the adsorption model is best for PSL. For insoluble particles, the experimental single parameter hygroscopicity is size dependent and decreases with an increasing diameter (Figure 2). This is because the core does not dissolve, and the total volume of the dry particle (Equation 2) does not fully participate in the water uptake. The particle surface of an insoluble particle is proportional to the square of the size and dominates the adsorption-based water activation behavior. However, the single parameter hygroscopicity is defined by the volume ratio of the solute and the solvent, which is proportional to the cube of the size. In adsorption driven growth, the core inactive particle volume enlarges when the particle size increases and as a result the perceived size dependence in the experimental hygroscopicity is exhibited. The FHH-AT model accounts for the core volume contribution to the water activation in the $B_{FHH}$ parameter. In the special case of PSL particles, $B_{FHH}$ happens to be 1(Table1), hence the $\kappa_{FHH,the}$ is directly the reciprocal of the dry size (Equation 8). $\kappa_{FHH,the}$ accurately describes the inactive core volume behavior, demonstrating a decrease of hygroscopicity with an increasing particle size. Larger particles still activate earlier than smaller particles. For example, 500nm particles activate earlier than 100 nm particles at 0.3% supersaturation due to a larger surface area which provides more active sites for adsorption. Moreover, FHH-AT hygroscopicity is also sensitive to the small differences in surface chemistry. The theoretical prediction for the CCN activation of PSL-NH$_2$ particles (the dashed line) is larger than PSL–COOH (dashed-dotted line), while the plain PSL is the lowest (solid line).

### 5. Summary and Implications





PSL are spherical, insoluble, non-porous, mono-dispersed particles and their surface can be modified to be hydrophilic or hydrophobic. The performance of the adsorption model is better than the traditional Köhler for PSL particles; experimentally derived $\kappa_{TK}$ predicts a higher hygroscopicity and $\kappa_{int}$ approaches zero. $\kappa_{FHK}$ is negative for the insoluble particles and is inconsistent with the growing droplets. The single
parameter hygroscopicity of the FHH-AT model replicates the small differences of the functionalized surface. $\kappa_{FHH,exp}$ and $\kappa_{FHH,the}$ are only within 5% differences. $\kappa_{FHH,the}$ decreases with an increasing particle size, demonstrating the inactivity of the inner core of larger particles.

Using a droplet growth model with additional degrees of freedom improves the droplet growth prediction of PSL. The FHH-AT model is typically applied to effectively water-insoluble particles and the
adsorption model agrees well with the experimental droplet growth data for hydrophobic and hydrophilic functionalized surfaces. This finding is consistent with the current body of work that highlights the bulk aerosol composition as a critical factor in aerosol water uptake. However, the addition of polar functional groups to the surface of the water-insoluble particle exhibits discernible differences in activation and suggests that for atmospheric insoluble aerosol (e.g., soot, mineral dust) the modified surface chemistry should not be
ignored.

Experiments show that $B_{FHH}$ was $\approx 1$ for pure and coated PSL particles. This implied that for the samples studied in this work, $B_{FHH}$ could be constrained to 1 to determine the $A_{FHH}$ and postulate the contribution of surface chemistry on the CCN activation. The method of constraining $B_{FHH}$ can distinguish both the surface modification and potential coatings when the $B_{FHH}$ of the core is known. If $B_{FHH}$ of the pure
core is known, only $A_{FHH}$ needs to be measured and accounted for to estimate water uptake ability. Furthermore, it was found that if both $A_{FHH}$ and $B_{FHH}$ are left unconstrained, the values of $A_{FHH}$ come out to be within 5% of the $A_{FHH}$ when $B_{FHH}$ is constrained. It is also important to note from Eq. (A10) that the $\kappa_{FHH}$ depends only on the surface properties of the compound ($A_{FHH}$) when $B_{FHH}$ is constrained to unity. This can imply that the hygroscopic properties of the compound will only depend on the hydrophilic or hydrophobic
properties of the functionalized surface regardless of the bulk properties if $B_{FHH}$ is constrained to unity. In other words, extensions of this work could potentially apply $A_{FHH}$ water-adsorption properties to similarly functionalized surfaces with different particle core compositions (i.e., varying $B_{FHH}$).

The findings presented here may be extended to atmospherically relevant insoluble particles that may be either coated or surface oxidized during different chemical processes. In ambient measurement, quite
rarely is the composition of the surface and core simultaneously known; single particle measurements are required to discern the individual composition and morphology. Thus, the singular $\kappa_{FHH}$ value provides an important utility. Regardless of whether the FHH-AT model is constrained, $\kappa_{FHH}$ reduces the the degrees of freedom of the model and can discern changes in surface chemistry. It should be noted that the extent to which the adsorption-driven droplet growth can be applied to increasingly hydrophilic aerosol is uncertain
but has been explored. Readers who are interested in the adsorption driven water uptake ability of partially soluble compounds are referred to a companion paper Gohil et al, *submitted*. In summary, water-insoluble aerosol can adsorb water and if their surfaces have been oxidized or functionalized with polar groups the aerosol can enhance their efficiency for water-uptake.

**Appendix**





A typical droplet growth model is expressed as,

$$S = a_w \cdot \exp\left(\frac{A}{D_p}\right), \tag{A.1}$$

where $S$ is saturation, $a_w$ is the water activity of the solution and $D_p$ is the diameter of the droplet. $A$ is a coefficient related to pure water droplet properties which is given as,

$$A = \frac{4 M_w \sigma_w}{RT \rho_w}, \tag{A.2}$$

Where $M_w$ is the molecular weight of water, $R$ is the gas constant, $T$ is the temperature and $\rho_w$ is the density of water. $\sigma_w$ is the surface tension of the droplet and is assumed to be the same as that of pure water.

If the solute is effectively insoluble, then the water activity term is expressed using the FHH-AT isotherm as,

$$a_w = \exp(-A_{FHH}\theta^{-B_{FHH}}), \tag{A.3}$$

where $\theta$ is the surface coverage term and is related to $a_w$ with the help of compound-specific empirical parameters ($A_{FHH}$, $B_{FHH}$). A simplified representation of $\theta$ is given as,

$$\theta = \frac{D_p - D_d}{2 D_w}, \tag{A.4}$$

where $D_p$ is the droplet size, $D_d$ is the dry particle size, and $D_w$ is the diameter of one water molecule. (A.3) can be equated with the $a_w$ parameterization defined in terms of the single hygroscopicity parameter ($\kappa$)

which is expressed as,

$$a_w = \exp(-A_{FHH} \cdot \theta^{-B_{FHH}}) = \left[1 + \kappa \cdot \frac{v_s}{v_w}\right]^{-1},$$
   (A.5)

Rearranging (A.5) provides the expression for $\kappa_{FHH}$ as,

$$\kappa_{FHH} = \frac{6 \theta D_w}{D_d}\left(\frac{1}{\exp(-A_{FHH}\theta^{-B_{FHH}})} - 1\right),$$
(A.6)

(A.6) is the function of the measured $D_d$ and $D_p$ derived corresponding the point of activation. (A.6) can be further simplified by making physically relevant mathematical assumptions for (A.5). The right-hand side of (A.5) can be simplified using the Taylor series expansion for an exponential function such that,

$$\exp(-A_{FHH}\theta^{-B\_FHH}) = 1 + (-A_{FHH}\theta^{-B\_FHH}) + (-A_{FHH}\theta^{-B_{FHH}})^2 + (-A_{FHH}\theta^{-B_{FHH}})^3 + \cdots,$$
(A.7)

Since $-A_{FHH}\theta^{-B_{FHH}} \ll 1$, (A.7) can be restated as,

$$\exp(-A_{FHH}\theta^{-B\_FHH}) \approx 1 + (-A_{FHH}\theta^{-B\_FHH}),$$
      (A.8)

The left-hand side of (A.5) can be simplified under the assumption that $v_w \gg v_s$ such that,


$$\left[1 + \kappa \cdot \frac{v_s}{v_w}\right]^{-1} \approx 1 - \kappa \cdot \frac{v_s}{v_w},$$ (A.9)

Combining (A.8) and (A.9) provides a reduced theoretical expression for $\kappa_{FHH}$,

$$\kappa_{FHH,th} = \frac{6D_w}{D_d} A_{FHH} \theta^{-B_{FHH}+1},$$
(A.10)

(A.10) contains $\theta$ defined as the point of activation such that $\theta = \theta_c$. $\theta_c$ is determined by taking the first
derivative of (A.1) and equating it to 0 to represent the point of activation such that,

$$\frac{dS}{dD_p} = \frac{d}{dD_p}\left(-A_{FHH} \cdot \left[\frac{D_p - D_d}{2 \cdot D_w}\right]^{-B_{FHH}} \cdot \exp\left(\frac{A}{D}\right)\right) = 0,$$
(A.11)

$$1 - \frac{2\theta_c D_w}{D_d} = \left(\frac{2AD_w}{A_{FHH}B_{FHH}D_d^2}\right)^{1/2} \theta_c^{\frac{B_{FHH}+1}{2}},$$ (A.12)

**Data Availability**

Additional Data is available upon request.

**Supplement Link:**

**Author Contributions:**

CNM conducted all experiments. CNM and AAA designed experiments. KG and CNM developed the single hygroscopicity parametrization. All authors contributed to the writing of the manuscript.

**Acknowledgements**

The authors would like to the thank funding and support from NSF-AGS:CHEM awards1708337 and 2003927.





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



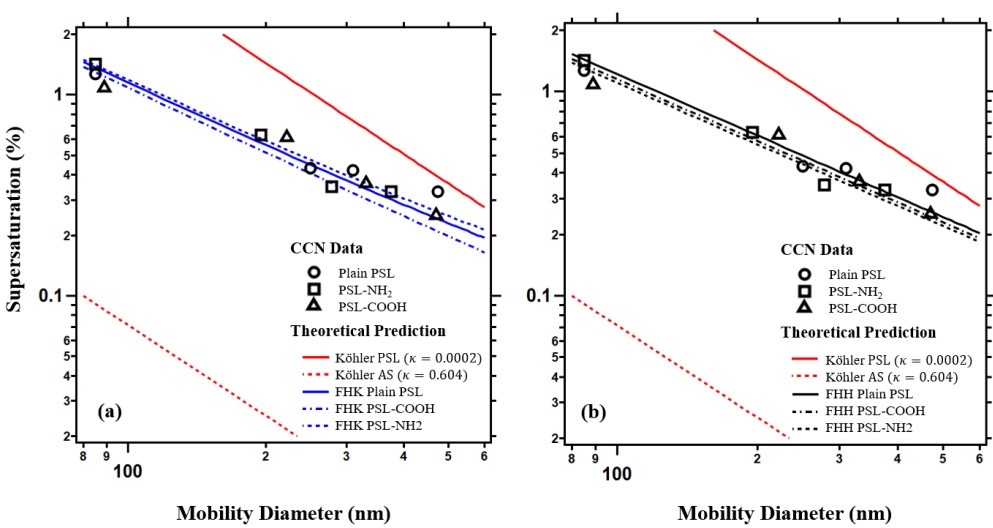


**Figure 1:** The $s_c$-$D_d$ data for different types of PSL. The dashed red line is the tradition Köhler prediction for ammonium sulfate ($\kappa = 0.604$). All types of PSL particles are more hygroscopic than the intrinsic hygroscopicity ($\kappa_{int} = 0.0002$, red solid line). (a) Blue lines show the prediction from FHK. (b) Black lines show the prediction from FHH-AT model, with $B_{FHH}$ =1.


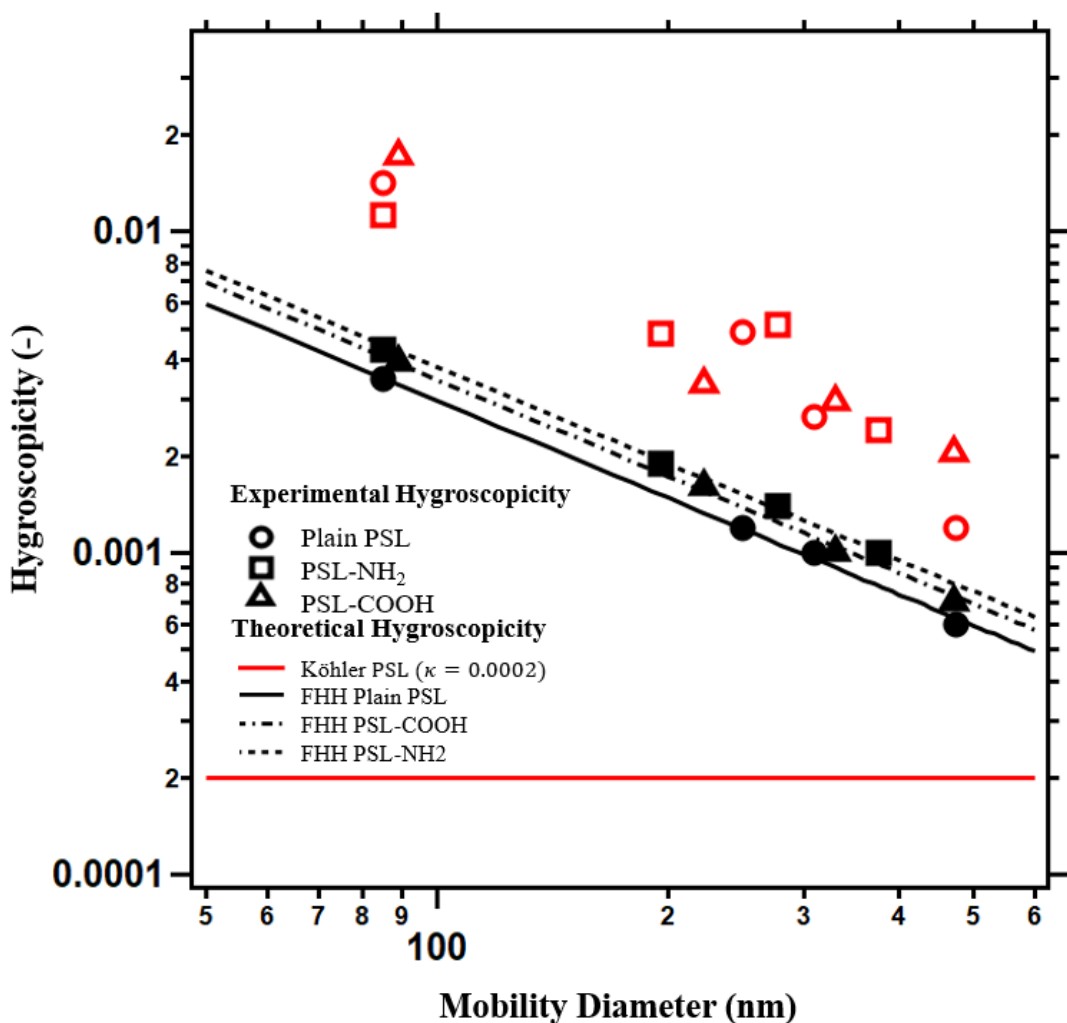

**Figure 2:** The single hygroscopicity parameter predicted from FHH-AT and traditional Köhler theory. Köhler theory from intrinsic properties (solid red line) predicts a constant hygroscopicity across particle sizes and hygroscopicity derived from experimental data (open symbols) shows size dependence. Hygroscopicity derived from FHH-AT model from theory (solid black lines) and from experimental data (black symbols) are size dependent and agree well. Hygroscopicity derived from FHH-AT model is also sensitive to surface chemistry functional.
