# Peer review of "A Single Parameter Hygroscopicity Model for Functionalized Insoluble Aerosol Surfaces"

_Atmospheric Chemistry and Physics, 2022_

## Author Comment (AC1)

We thank the reviewers for their comments and suggestions and have provided point by point responses.

**Referee 1:**

General comments:

Authors proposed a new single parameter hygroscopicity representation for insoluble aerosol surfaces, and have done comparisons with traditional TK or FHK models. The proposed model might be extended to atmospherically relevant insoluble particles and findings of this search reveal that water-insoluble aerosol can adsorb water if their surfaces have been oxidized or functionalized with polar groups, thus of importance to atmospheric aerosol research. I only have some minor and specific comments.

1.    The logic of the introduction is not clear, and hard to follow. For example, a lot of discussions about the FHK model in the results part, but very few descriptions in the introduction. In my opinion, both the FHK and TK should be introduced before the discussions of FHH-AT.

We have revised the introduction to improve flow and clarity. Specifically, our primary intent is to discuss the importance of the adsorption model (FHH). TK is the simplest and most widely used model and is discussed for comparison.   We thought twice about including FHK.  FHK is less known but should be applied to water-soluble polymers. PSL is a polymer but not water-soluble and as we show later on, should not be used on PSL particles.  Hence, a too  lengthy discussion of FHK or TK will distract from the primary FHH and single hygroscopicity parameter message.   We use FHK and TK, two water-soluble models, compared with the FHH-AT to show the importance of applying the right theory to specific compounds. In the new revision of the article, we have added sentences as requested.

The text has revised sentences and an additional paragraph has been added, and is as follows:

"Flory Huggins Köhler (FHK)(Petters et al., 2009) is one example of a droplet growth model specifically applied to the water soluble polymers.  FHK has been shown to work well for long-chained polymers such as gelatin, polyethylene glycol and polylactic acid (Mao et al., 2021; Petters et al., 2006, 2009). It uses a one fitting parameter that describes solvation and most recently was incorporated into a single-parameter hygroscopicity term that describes the water-uptake of water-soluble aerosol (Mao et al., 2021)."

2.    The TK model directly gives the relationship between aerosol growth factor and relative humidity (saturation ratio), suggest authors also present a direct formula that links RH, Dd(dry diameter) , Dw (wet diameter) and the single hygroscopicity parameter.

It should be noted that we do not show results from the sub-saturated regime in this paper. PSL does not grow in the subsaturated regime.  Thus, equations related to growth factor and RH are not used or presented in this paper.  However,  FHH does extend into the subsaturated regimes, and thus we refer the author to relevant papers that show sub-saturated FHH work.  Furthermore,  we encourage the reviewers to explore our companion paper Gohil et al, 2022 (ACPD) that is a hybrid activation adsorption and solubility model that is more robust,  can be applied to subsaturated measurements, and has direct equations for the subsaturated environment.

We provide the HTDMA result (RH=91%) of the PSL around 300 nm and show that PSL particles are actually hydrophobic. The growth factor for all three particles are all around 1, showing that the particles do not grow under sub-saturation regime.

[Figure]

HTDMA-91%-300nm

|  | -NH2 | -COOH | -plain |
|---|---|---|---|
| Dd | 278 | 331 | 310 |
| HTDMA | 294.1±22.86128 | 340.37±20.97137 | 326.39±24.78197 |
| HGF | 1.057914 | 1.028308 | 1.052871 |

**Specific Comments:**

L37-38, "for water-soluble particles……, TK can accurately predict their water uptake behavior", I am not sure whether use "accurately predict" is correct. Even the aerosol particle is water soluble, the performance of TK still depends highly on the solubility 1.

The word "accurately" has been deleted. The sentence now reads:
"For water-soluble particles like inorganic ammonium sulfate (Rose et al., 2008) and sucrose (Dawson et al., 2020; Gohil and Asa-Awuku, 2022), TK can predict their water uptake behavior"

L40 "partially water soluble corresponding to very small solubility" or has other physical understanding?

The value of solubility was added to the text. The sentence now reads:

"However, TK does not work so well for atmospherically relevant and abundant particles that are partially water soluble or water insoluble, less than a concentration of $5 \times 10^{-4}$ (Kumar et al., 2009; Petters and Kreidenweis, 2008; Tang et al., 2016). Thus, alternative droplet growth models for the partial and insoluble particles are needed. "

L44, BET does not appear again in the following, is the abbreviation necessary?

Abbreviation deleted. The sentence now reads:
"Brunauer, Emmett, and Teller adsorption isotherm models are typically applied for multilayer adsorption analysis of water uptake on clays (Hatch et al., 2012) and fly ash (Navea et al., 2017)."

L95-96, should use TK and FHK?

Changed. The sentence now reads:

"Two single parameter hygroscopicity representations have been previously derived using TK (Petters and Kreidenweis, 2007) and FHK (Mao et al., 2021) assumptions."

L119  flowrate of L/min is better

Corrected. The sentence now reads:

"The PSL particles are then passed through a Condensation Particle Counter (CPC, TSI 3776) with a flow rate of 0.3 L min$^{-1}$ and a CCNC. "

L159 the van't Hoff factor is missing

Corrected.

The sentence now reads:

"$\kappa\_int = v\ (M\_w\ \rho\_s)/(M\_s\ \rho\_w\ )$   (Sullivan et al., 2009). Where M_w is the molecular weight of water; M_s is the molecular weight of the dry particle; $\rho$_w is the density of water; $\rho$_s is the density of the dry particle; v is the van't Hoff coefficient that is one."

L227 change "and" to ";" before AFHH?

Corrected. The sentence now reads:

"The fit results for plain type PSL estimate $A_{FHH}$ = 0.17 and $B_{FHH}$ = 0.99; $A_{FHH}$ is 0.3 and $B_{FHH}$ is 1.08 for carboxyl functional group modified PSL; $A_{FHH}$ is 0.11 and $B_{FHH}$ is 0.83 for amine functional group modified PSL."

L258 "derived is", delete "derived"

Corrected. The sentence now reads:

"the derived hygroscopicity is problematic and nonsensical (therefore not shown)."

---

## Author Comment (AC2)

We thank the reviewers for their comments and suggestions and have provided point by point responses.

**Referee 2**

The manuscript by Mao et al. quantified CCN activity of plain and surface-treated polystyrene latex (PSL) particles. The data were interpreted using the Köhler theory by considering a few different types of water-particle interactions. The authors claim that the result of the present study can be useful for interpreting CCN activation of atmospheric insoluble particles. The manuscript is well organized. It is easy to follow the story. However, I have some concerns about the data quality.

Major comments

Commercially available solutions for PSL particles typically contain surfactants for avoiding coagulation (Kidd et al., 2014). The manuscript provides no descriptions about potential influences of surfactants on the data. The reviewer checked the website of the manufacturer of PSL particles for the present study to search for the corresponding information. It seems that the manufacturer adds some additional chemical species to stabilize the PSL solutions (if I understand it correctly). As the manuscript only provides CCN spectra in the supplement as results, it is difficult to judge the potential artifacts on the data. The reviewer is not sure how it influences the conclusion of the study. As the manuscript focuses on CCN activity of PSL particles, quantitative information on this point would be needed for evaluating the data quality. At the current moment, the reviewer is unable to judge the technical validity of the study due to the lack of this information.

We do not believe that the additive is influencing the results presented here. To address the concerns we conducted additional experiments to provide evidence to the robustness of the data presented. Specifically, We took careful consideration in selecting particles and have provided the following information to address concerns.

1. We have emphasized in the text that the reported sizes of the particles are based on the measured electrical mobility diameter. For each PSL solution and after the atomization process, we first did a scan of all aerosols within the size range of 6 to 386 nm. The reported electrical mobility diameters are based on the peak determined in these scans. Furthermore, we believe the surfactants to mainly form peaks at 20~100nm. (see the figure below). The orange line is the theoretical fit of surfactant particles. The PSL sizes selected for CCN activation in the manuscript are larger than the surfactant size distribution.

2. To provide additional evidence of the different compositions at different peaks, we measured the CCN activity of surfactant peak (size distribution) of particles with two methods.    In the first method, we scanned through particle sizes in this range and measured the CCN activity at 0.8% supersaturation and found a critical diameter of 40 nm (see figure below).  Using Kohler Theory we estimate that the aerosols within this size range have an estimated Kappa- hygroscopicity of 0.28±0.05.  The TK kappa value for this peak is 10 to 100 times larger than the measured TK hygroscopicity of PSL ranging from 0.002 to 0.02 (see Fig 2 in the manuscript).    We also measured CCN activity with a second method.  That is, we selected 60 nm particles and found a critical supersaturation around 0.5% by varying instrument supersaturation.  With this method, we estimate the kappa hygroscopicity to be  0.4.

3. The hygroscopicity is consistent with the composition of a hydrophilic surfactant. The Material Safety Data Sheet (MSDS) suggests tween20 is the surfactant used in the solution.   The behavior of the hydrophilic Tween20 is vastly different from the hydrophobic PSL particles.  Compositions that readily dissolve in water, like hydrophilic salts and surfactants components do not exhibit a hygroscopicity dependence with diameter. Figure 2 in the manuscript shows that the TK hygroscopicity of the PSL peak is size dependent.

4. The reviewer suggests that there maybe contamination of the surfactant on the size selected PSL particle.  We can do a quick calculation of the impact of a mixed PSL and surfactant aerosol. Let's say there is a 5% by volume mixture of Tweek20. Assuming a simple ZSR, a 5% or more influence of surfactant would result in kappa values greater than 0.017.  In figure 2, the majority of PSL TK measurements are less than 0.01, indicating that any contamination is much smaller than 5% and negligible.  The measured hygroscopicity is size dependent and must be mainly due to non-hygroscopic PSL spheres.

5. If indeed there is some surfactant (less than 5% of the volume of the mixture), the same surfactant is used in all samples produced by the manufacturer.  And thus any changes observed (even minor) can be attributed to the differences in PSL structure and composition.

6. Again, we do not believe that the size selected  PSL aerosol contains surfactant. The PSL hygroscopicity measured is much smaller than the pure surfactant.  The raw data (Figure S1) shows only one sigmoid curve, indicating a singular composition at the selected particle size.

[Figure]

SMPS of a 300nm PSL sample. The orange line is the Guassian fit for the small surfactant particles.

[Figure]

SMCA data PSL(300 nm, plain) sample at 0.8% supersaturation.

Size selection of PSL particles by the DMA should be conducted more carefully. Most of (e.g., NaCl, (NH4)2SO4) atomizer-generated particles have number size-distributions that are significantly broader than the DMA transfer function. In these cases, the setting diameter for the DMA and the mode diameter of the selected particles agree. However, standard PSL particles typically have narrow size distributions. The widths of the distributions are comparable to that of DMA transfer functions in many cases. So, it would be ideal to measure the size distributions of PSL particles using the DMA for matching the mode diameters for PSL and DMA transfer function. The manuscript describes that the authors measured the PSL particles using the DMA and CPC. However, it is not clear how the DMA diameter was finally set.

In the revised manuscript we have clarified the method for PSL size selection. In fact, in earlier drafts we had included this information and then took it out. In the revised manuscript, we emphasize that we actually double checked sizes. Furthemore, we have provided size information data. We first conducted scans over a size range between 6 nm to 386 nm. An example figure of a scan is shown below. Rather than use a tandem set-up which requires a second system to be calibrated, we use the same system to size-select the aerosol (PSL and ammonium sulfate). It is important to note that the DMA system is the same as used for calibration of the instrument supersaturation. It should also be noted that the electrical mobility diameter measured is within 10% of the manufacturer's reported aerosol diameter estimated with dynamic light scattering. Although true that the DMA transfer function is wider at larger aerosol sizes, the uncertainty in this width still indicates that the PSL will have a very low hygroscopicity (similar to that reported here). Indeed, it is reported that other sizing instrumentation (beyond that used here) maybe more suitable for larger aerosol widths (Gohil and Asa-Awuku, 2022). This is stated in Section 2.1.

As shown in the abstract, the authors concluded that the plain PSL particles are less hygroscopic than functionalized PSL particles. Figure S1 shows the experimental data for the study. I agree with the statement for 100 nm particles. However, the data for 200 nm particles exhibit an opposite trend. A clear explanation would be needed.

As stated in the caption for figure S1. "Larger particles will activate earlier and therefore should not be directly compared". As previously mentioned the sizes of particles purchased were re-measured with the DMA. Thus ~200 sized particles of PSL were, 250nm (plain) 223 nm (PSL-COOH), and 195 nm (PSL-NH2). Larger particles provide more surface area for water to condense, and hence the corresponding activation data for plain PSL with larger sizes will appear to be more active than PSL-COOH and PSL-NH2. We thought about this deeply. Including the raw data activation figure of

different sized particles is misleading. We have indicated that for the supplemental graphs, readers should refer to the measured electrical mobility diameters provided in the main manuscript. The best way to do a direct comparison of the activation is with either sc-dd pairs or with hygroscopicity parameters (hence Figures 1 and Figures 2). In this paper, we focus on the surface property contribution to the hygroscopicity but not the particle size. Hence we developed the hygroscopicity based on the adsorption theory.

Minor comments

L114 Silicone dryers

Silicagel dryers?

Corrected. The sentence now read:

"Wet droplets were then passed through two silica gel dryers and the relative humidity was 5% after passing through the dryers."

L207

I agree that the data suggest the PSL particles for the present study look like hygroscopic based on the data. Does it agree with identified bulk property of PSL?

Overall the PSL are not very hygroscopic. However Yes, the findings are consistent with what we know of PSL; the bulk property of plain PSL is hydrophobic and water-insoluble. The surface of the PSL can be modified either hydrophilic or hydrophobic.

---

## Author Response (AR2)

Thank you for reconsidering this work for publication.

In response to the Editors comment,
Please include the surfactants discussion from Anonymous Referee #3 in the main text, and then the manuscript is ready to be published

We have added the following text to Section 2.1

It should be noted that the PSL solution from Lab 261® contains Tween 20 surfactant; the surfactant is added to prevent coagulation. The atomized surfactant particles are much smaller than the PSL particles and form aerosol less than 100 nm. Thus the majority of surfactant particles are excluded from the size-selected CCN measurement of the large PSL particles. The raw CCN data (Fig. S1) shows only one sigmoid curve, indicating a uniform composition at the selected particle size. Moreover, the apparent hygroscopicity of surfactant particles, typically a hydrophilic polymer, is much larger than an insoluble polymer like PSL. The measured TK hygroscopicity of Tween 20 is ~ 0.28±0.05, while the measured TK hygroscopicity of the PSL particles varies from 0.002~0.02. (see Fig. 2.) and thus the influence of the surfactant on the size-selected PSL aerosol is assumed negligible.